# Spatio-Temporal Analysis and Health Risk Assessment of Heavy Metals in Water from the Fuhe River, South China

Xiaojuan Guo [1], Yilong Xiao [2], Lanzhi Zhao [1], Tao Yang [1], Chun Tang [3], Wei Luo [4], Cheng Huang [3] and Fangwen Zheng [3,*]

1 China Railway Water Conservancy & Hydropower Planning and Design Group Co., Ltd., Nanchang 330029, China
2 Jiangxi Provincial Water Investment Engineering Consulting Group Co., Ltd., Nanchang 330029, China
3 School of Hydraulic and Ecological Engineering, Nanchang Institute of Technology, Nanchang 330029, China
4 Power China Northwest Engineering Corporation Limited, Xi'an 710065, China
* Correspondence: zhengfangwen82@163.com

**Abstract:** With rapid developments in society and economy, the concentrations of heavy metals in surface water in South China have increased significantly, which poses a serious threat to the regional water security. In this study, the Fuhe River watershed in South China was selected as the study area to analyze physicochemical characteristics and heavy metal (Cu, Zn, Pb, Cd and Cr) concentrations in river water in the dry and rainy seasons, in 2019, with the purpose of exploring their spatial–temporal variations and main influences and assessing the potential health risks of heavy metals. The pH value of river water varied from 5.82 to 7.97, and it fluctuated less overall in the dry season, but it oscillated greatly in the rainy season and was lower, especially in the lower reach of the Fuhe River. The electrical conductivity (EC) value changed between 33 μS/cm and 128 μS/cm and increased and fluctuated along the river flow in the two periods. The concentrations of Cd, Cr, Cu, Pb, and Zn in river water showed obvious differences between the two periods. The concentrations of Cd and Cu were lower with the ranges from 0.001 μg/L to 0.67 μg/L and from 1 μg/L to 12 μg/L, respectively, in the dry season than in the rainy season, while there were inverse cases for other heavy metals. Along the river flow, the Cr concentration was stable, whereas other heavy metals showed increasing trends. It was noted that the concentrations of heavy metals in the Fuhe River were, on average, lower than the Chinese drinking standard values, with the concentration of Pb in the dry season significantly higher than the drinking standard value of the World Health Organization (WHO). Principal component analysis and correlation analysis showed that rock weathering and anthropogenic inputs were the main controlling factors of Cu and Zn in the Fuhe River, and human activities were mainly responsible for Pb, Cr, and Cd concentrations. The health risk assessment results showed that the non-carcinogenic risk ($HQ_{ingestion}$) value of Pb was greater than 1 in most sampling points in the middle and lower reaches in the dry season, suggesting a significant non-carcinogenic risk for adults and children by direct ingestion. The minimum carcinogenic risk ($CR_{ingestion}$) value of Cr was more than $10^{-4}$ in the rainy and dry seasons, and the $CR_{ingestion}$ value of Cd in some sampling points was more than $10^{-4}$ in the rainy season, indicating significant cancer risks to adults and children. For areas with significant pollution and health risks in the Fuhe River watershed, it is urgent to strengthen the controls of industrial, agricultural, and urban wastewater discharge.

**Keywords:** Fuhe River watershed; surface water; heavy metal concentrations; controlling factor; non-carcinogenic and carcinogenic risks

## 1. Introduction

Rivers are an important part of the terrestrial ecosystem, which provide abundant water resources for the sustainable development of human society and ecological environments. With the acceleration of industrialization and urbanization, based on the impact of

natural sources, unreasonable human activities strengthen pollution risks of heavy metals in rivers, which pose direct harms to human health [1–3]. Previous studies have shown that people (including adults and children) exposed to heavy metals (e.g., Cu, Zn, Cd, Pb, and Cr), through direct ingestion and dermal absorption, usually face cancer or non-cancer risks, such as lung cancer, nervous system diseases, and intellectual disability [4]. Natural sources, such as rock weathering, soil erosion, atmospheric deposition, and microbial degradation, usually contribute less to heavy metals in rivers [5,6]. Anthropogenic sources mainly include municipal discharge, agricultural fertilizer, and industrial pollution, and they are considered to be direct causes for the pollution of heavy metals in rivers [7,8]. In recent decades, the heavy metal pollution of rivers has been a research hotspot at home and abroad, and authors have carried out a lot of work on this issue, involving the form distribution, migration, release and enrichment, pollution, and risk assessments of heavy metals in river (water) sediments [9–11]. Prior findings are of theoretical value for understanding the geochemical cycle of heavy metals in rivers and have practical significance for guiding river pollution control.

Compared with foreign countries, studies on heavy metals in rivers lag behind slightly in China. However, with the advancement of national economy and people's yearning for a better life, the studies of this field have recently made great progress in China [7,11–17]. For example, using multivariate statistical analysis, it was pointed out that river water in the upper reach of the Han River was mainly polluted by As, Cd, Pb, Sb, and Se, which were related to man-made discharge [7]. On the basis of investigating the transport characteristics of heavy metals in surface water of the Poyang Lake Basin, it was found that there was the most serious water pollution in the rainy season [12]. Some authors emphasized that the concentrations of Cu, Cd, Zn, and Pb in the sediments of the middle and lower reaches of the Yangtze River led to the basin ecosystem being at risk of pollution [13]. Most previous studies focused on large rivers and lakes, but environmental issues of heavy metals in small river watersheds were not understood enough [11,14,15]. In the economically developed and heavily polluted southern regions of China, small river watersheds are well developed, and they are also the headwater areas of large rivers. In addition, industrial and mining enterprises and agricultural planting in small river watersheds are currently the blind and difficult spots of the government's environmental supervision [16,17]. Therefore, it is necessary and urgent to investigate environmental issues of heavy metals in small river watersheds.

The Fuhe River watershed is typically a small watershed in South China, which is located in the middle-lower reach of the Yangtze River. The Fuhe River, which eventually flows into the largest freshwater lake in China, Poyang Lake, is an important water source for drinking water and irrigation for Nanchang City and its surrounding areas in Jiangxi Province. In recent years, with the rapid development of industry and agriculture in the watershed, the heavy metal pollution in the Nanchang section of the Fuhe River has become increasingly serious [18]. Until now, some authors investigated the distribution and influencing factors of heavy metals in the Fuhe River, but most of them were concentrated in some sections of this river [19,20], and less consideration was given to the spatial distribution and seasonal differences of the whole watershed, restricting in-depth understanding of the distribution law and impact mechanism of heavy metals in the small watershed to some extent. The potential health risk assessment of water bodies in the watershed was also scarce. For the above-mentioned reasons, in this study, the concentrations of heavy metals were investigated in the Fuhe River during the rainy and dry seasons. Our aims were to find out spatial and seasonal differences of heavy metals in the Fuhe River water, to explore the main factors affecting heavy metals, and to assess the potential health risks of heavy metals. This finding is expected to provide a scientific basis for decision-making on water resource management and water pollution control of the Fuhe River watershed and similar small watersheds.

## 2. Materials and Methods

### 2.1. Study Area

The Fuhe River is the second largest river, with a length of about 350 km in Jiangxi Province. It originates from the Linghua Peak (991 m above sea level) in Guangchang County, and it finally flows into the Poyang Lake (Figure 1). It is an important water source for drinking water and irrigation in Nanchang City and the surrounding regions (Figure 1). The total area of the Fuhe River watershed (115°36′~117°10′ E, 26°30′~28°20′ N) is about 15,832 km². The upper reach is from the headwater of the Fuhe River to Nancheng County, and the middle reach is from Nancheng to Chongren County, and the lower reach is below Chongren County (Figure 1). Geomorphologically, the middle and upper reaches of the watershed are dominated by hills, while the lower reach is controlled by gentle alluvial plains. Lithologically, the upper reach of the watershed consists of metamorphic and magmatic rocks, the middle reach comprises sedimentary rocks, and the lower reach is composed of loose layers, such as sand, gravel, and clay (Figure 1). Overall, the Fuhe River watershed is a silicate rock-predominated area, and weathering of the watershed has been demonstrated to be dominated by silicate weathering [21]. Climatically, there is a subtropical monsoon climate, with four distinct seasons in the watershed. The continental cold air mass predominates during the cold and dry season. During the warm and rainy season, heavy rains occur frequently due to the intersection of cold and warm air masses. In this watershed, the average annual precipitation is 1500~2000 mm/a, and the average annual evaporation is 1343~1600 mm/a [20,22]. The average annual runoff is $159 \times 10^8$ m³/a, and the runoff during the rainy season (April to September) accounts for 67.8~74.1% of the rain for the whole year. The dry season lasts from October to next March [23]. Agricultural planting is dominant in the upper reach of the Fuhe River watershed, and the middle and lower reaches are characterized by industrial and urban activities. There are some transportation routes, which are usually more than 100 m far from riverbank in the upper and middle reaches of the Fuhe River watershed, where rural areas predominate with small populations. In the lower reaches, where there are many towns, vehicle emissions probably have certain effects on river water quality.

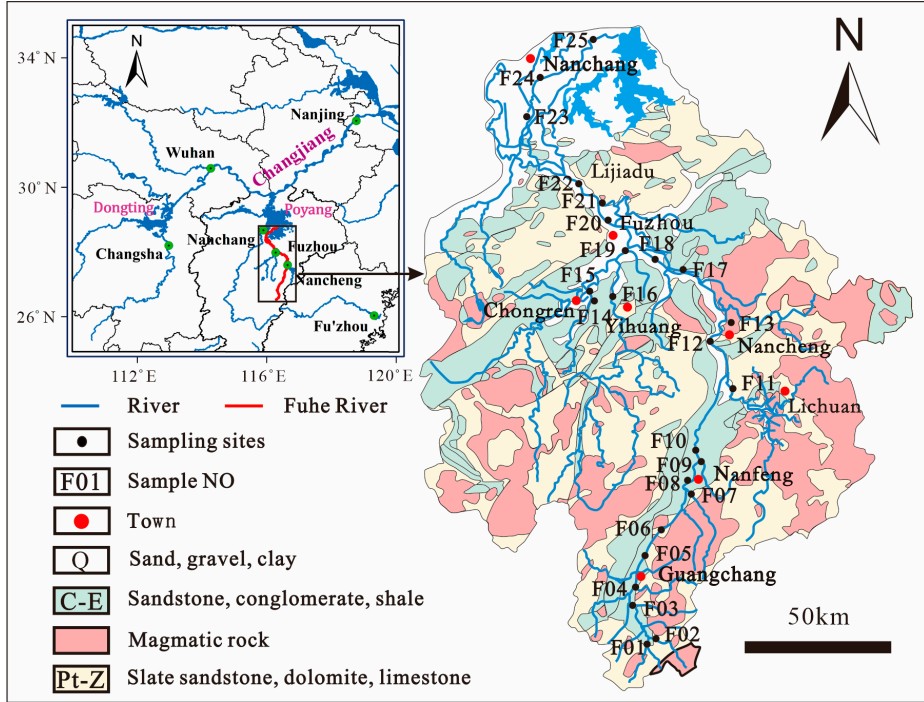

**Figure 1.** Simplified map showing the lithology and sampling sites in the Fuhe River watershed.

## 2.2. Sampling and Analysis

Sample collections were conducted once in the main stream and tributaries of the Fuhe River, respectively, in January (dry season) and August (rainy season), 2019, with a total of 50 river water samples (Figure 1). January and August are usually characterized by dry and rainy climates, respectively, in Jiangxi [24], so the two months were selected as the sampling periods in this study. Each sample, comprising about 500 mL, was taken at a depth of 50 cm below the water surface in the flowing waterbody and, thus, can reflect the local hydrology and environment without sample replication. The pH and electrical conductivity (EC) of river water were measured in the field using a multi-parameter water quality analyzer (HannaH198292G, Italy). The river water sample was filtered by a 0.45 μM cellulose acetate filter membrane and then acidified to pH < 2 with ultrapure hydrochloric acid. In the laboratory, concentrations of heavy metals, such as Zn, Cu, Cd, Pb, and Cr in the samples, were determined by inductively coupled plasma mass spectrometry (Agilent 7700 ICP-MS). Previous studies showed that the concentrations of Zn, Cu, Cd, Pb, and Cr were significantly greater in the nearby rivers [25,26], so these heavy metals were considered as study targets in this study. Reagents and procedural blanks were determined in parallel to the sample treatment using identical procedures. Each calibration curve was evaluated by analyses of quality control standards before, during, and after the analyses of a set of samples. The analytical precision was within 5%, and the analytical accuracy was less than 10% for the heavy metals.

## 2.3. Data Processing

### 2.3.1. Statistical Methods

Principal component analysis (PCA), together with correlation analysis (CA) and hierarchical cluster analysis, has proven to be a useful multivariate statistical technique for disclosing the origins of heavy metal contamination [6,27–29]. In this study, Pearson's correlation analysis was conducted to analyze the correlations between the variables because the normality distributions of their data were tested and confirmed. PCA is often used in data reduction to identify common factors (principal components and PCs) that explain most of the variance observed in a large number of manifest variables. The reduced, new set of orthogonal (non-corrected) PCs by PCA is arranged in decreasing order of merit. PCA was made with varimax rotation of standardized component loadings for maximizing the variation among the variables under each factor [30] (Abdi and Williams, 2010). The eigenvalue for the factor represents the strength of variance for the interpretive variables, and only eigenvalues ≥ 1.0 were considered in this study. The hierarchical cluster analysis (HCA) was used to classify similar samples, with built-in algorithms, and it has been widely used in many fields [31] (Chen et al., 2020). The three analyses were all performed using SPSS software, version 19.0. The pH, EC, and heavy metal concentrations were statistically expressed as the box–whisker plots using OriginPro2021. The box–whisker plot shows the minimum, first quartile, median, third quartile, and maximum in a set of data, and it can thus indicate the central location and dispersion range of data distribution.

### 2.3.2. Human Health Risk Assessment (HHRA)

The health risk assessment was performed with the model recommended by USEPA for river water [32]. This method relates human health with environmental pollution, and it quantitatively describes the risk of human health hazards caused by pollution [33]. The non-carcinogenic risk assessment was carried out for Zn, Cu, Cd, Pb, and Cr, and the carcinogenic risk assessment was made for Cd and Cr, which have potential carcinogenic effects, but not for other heavy metals, which are not included in the carcinogenic risk (CR) calculation [34,35]. Daily average intake dose is a key parameter used to calculate non-carcinogenic and carcinogenic risk parameters, which were calculated by Formulas (1) and (2), respectively.

$$CDI_{ingestion} = \frac{C_i \times IR \times EF \times ED}{BW \times AT} \tag{1}$$

$$CDI_{dermal} = \frac{C_i \times SA \times AF \times ABS_d \times ET \times EF \times ED \times CF}{BW \times AT} \tag{2}$$

where $CDI_{ingestion}$ and $CDI_{dermal}$ refer to the long-term daily mean radial intake and skin exposure doses [mg/(kg·d)], respectively, and $C_i$ means the measured concentration of heavy metal $i$, and other parameters were listed in Table 1.

**Table 1.** General parameters of the health risk assessment for heavy metals in river water.

| Parameter | Abbreviation | Unit | Mean Value | |
|---|---|---|---|---|
| | | | Adult | Child |
| Ingestion rate | IR | L·day$^{-1}$ | 2.65 [a] | 0.78 [b] |
| Exposure frequency | EF | days·year$^{-1}$ | 365 [b] | 365 [b] |
| Exposure duration | ED | year | 70 [c] | 6 [c] |
| Skin-surface area | SA | cm$^2$ | 16,000 [a] | 5700 [d] |
| Adherence factor | AF | cm·h$^{-1}$ | 0.07 [d] | 0.07 [d] |
| Dermal absorption factor | ABS [d] | | 0.03 [d] | 0.03 [d] |
| Exposure time | ET | h·day$^{-1}$ | 0.6 [e] | 0.6 [e] |
| Conversion factor | CF | L·cm$^{-3}$ | 10$^{-6}$ [c] | 10$^{-6}$ [c] |
| Body weight | BW | Kg | 60.5 [a] | 15 [f] |
| Average time | AT | day | ED × 365 | ED × 365 |

Note: [a] Exposure factors handbook for the Chinese population [33]. [b] Risk Assessment Guidance for superfund I: Human Health Evaluation Manual [32]. [c] Supplemental Guidance for Developing Soil Screening Levels for Superfund Sites, Appendix D—dispersion factors calculation [36]. [d] Exposure Factors Handbook [37]. [e] Preliminary risk assessment of trace metal pollution in surface water from Yangtze River in Nanjing section, China [38]. [f] Human health evaluation manual, supplemental guidance, standard default exposure factors [39].

The non-carcinogenic risk value (HQ) of human health was calculated according to Formulas (3) and (4), which can assess the non-carcinogenic risk hazards of heavy metals to human health. When HQ > 1, there is a health risk to human beings, otherwise, there is little or no health risk [28].

$$HQ_{ingestion} = \frac{CDI_{ingestion}}{RfD_{ingestion}} \tag{3}$$

$$HQ_{dermal} = \frac{CDI_{dermal}}{RfD_{dermal}} \tag{4}$$

where $RfD_{ingestion}$ and $RfD_{dermal}$ represent the reference doses for direct ingestion and dermal exposures, respectively. $RfD_{ingestion}$ is determined as 300 µg/kg/d for Zn, it is 40 µg/kg/d for Cu, it is 1.4 µg/kg/d for Pb, it is 0.5 µg/kg/d for Cd, it is 3 µg/kg/d for Cr, $RfD_{dermal}$ is 60 µg/kg/d for Zn, it is 12 µg/kg/d for Cu, it is 0.42 µg/kg/d for Pb, it is 0.005 µg/kg/d for Cd, and it is 0.015 µg/kg/d for Cr [28].

The carcinogenic risk value (CR) of human health may assess the carcinogenic risk hazards of heavy metals to human health. The risk values may be calculated according to Formulas (5) and (6) if CR < 0.01, otherwise, they are obtained according to Formulas (7) and (8). The acceptable CI value range of USEPA (1989) for human health is $10^{-6} \sim 10^{-4}$. The CI value of $<10^{-6}$ indicates that the carcinogenic risk is negligible, while the CI value of $>10^{-4}$ means that the carcinogenic risk is significant [28,29].

$$CR_{ingestion} = SF \times CDI_{ingestion} \tag{5}$$

$$CR_{dermal} = SF \times CDI_{dermal} \tag{6}$$

$$CR_{ingestion} = 1 - \exp\left(-CDI_{ingestionl} \times SF\right) \tag{7}$$

$$CR_{dermal} = 1 - \exp(-CDI_{dermal} \times SF) \tag{8}$$

$$CI = CR_{ingestion} + CR_{dermal} \tag{9}$$

where *SF* is the carcinogenic slope factor, $SF_{Cd}$ is determined as 0.0061 kg·d/μg, and $SF_{Cr}$ is 0.041 kg·d/μg [28,29].

## 3. Results and Discussion

### 3.1. Distribution Characteristics of pH, EC, and Heavy Metals in River Water

The pH value of river water changed between 7.23 and 7.97 in the dry season, with an average value of 7.64, while it varied from 5.82 to 7.81 in the rainy season, with an average value of 7.14 (Figure 2). The pH value of river water was within the range of drinking water standards of the WHO and China during both seasons (Table 2). The pH value of river water was relatively lower in the rainy season and showed a great variation along river flow (Figure 3a). In the dry season, the pH value of river water was relatively lower in the upper reach, but it was higher in the lower reach and showed a narrow fluctuation along the river flow (Figure 3a). Compared with other rivers within the Poyang Lake Basin, the pH value of the Fuhe River was close to that of the Le'an River in the rainy season and slightly higher than that of the Le'an River in the dry season (Table 2). Compared with the Yangtze River and Yellow River, the pH of the Fu River was relatively lower [40,41]. The electrical conductivity (EC) of river water in the dry season varied from 35 μS/cm to 128 μS/cm, with an average value of 59.8 μS/cm, slightly higher than that in the rainy season, which may be related to the dilution effect of the Fu River runoff during the rainy season (Figure 3b). In the two seasons, the EC value of river water showed similar spatial variations: from upstream to downstream, the EC value increased and oscillated (Figure 3b). Compared to the Yangtze River and the Yellow River, the EC value of the Fu River was relatively lower [40,41]. Compared with the Le'an River, the EC value of the Fuhe River was significantly lower (Table 2). The variation law of pH and EC values along the Fuhe River was closely related to industrial, agricultural, and urban activities. Especially, in the rainy season, the pH decrease and the EC increase in river water in the lower reach of the Fu River were directly related to frequent human activities.

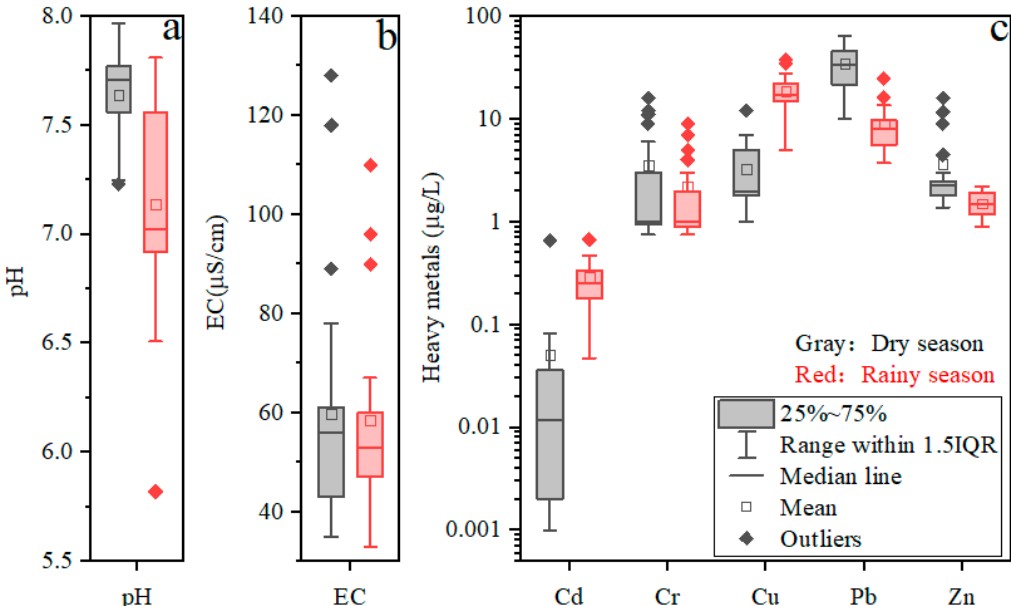

**Figure 2.** The statistical summary of (**a**) pH, (**b**) EC values, and (**c**) heavy metal concentrations of the Fuhe River water.

**Table 2.** The mean values of pH, EC (μS/cm), and heavy metal concentrations (μg/L) in the Fuhe River and the surrounding rivers.

| Rivers | Period | pH | EC | Cu | Zn | Pb | Cd | Cr | Reference |
|---|---|---|---|---|---|---|---|---|---|
| Fuhe River, China | Dry season | 7.64 | 59.8 | 3.12 | 3.60 | 34.94 | 0.051 | 3.66 | This study |
| | Rainy season | 7.14 | 58.48 | 17.64 | 1.43 | 8.76 | 0.286 | 2.09 | This study |
| Ganjiang River, China | Dry season | - | - | 1.98 | 4.32 | 1.42 | - | 1.71 | [42] |
| | Rainy season | - | - | 4.81 | 18.16 | 5.36 | - | 4.52 | [42] |
| Xiuhe River, China | Dry season | - | - | 6.15 | 5.81 | 2.71 | - | 2.53 | [42] |
| | Rainy season | - | - | 4.65 | 18.53 | 6.45 | - | 6.15 | [42] |
| Xinjiang River, China | Dry season | - | - | 5.81 | 30.05 | 1.89 | - | 4.55 | [42] |
| | Rainy season | - | - | 8.82 | 17.89 | 5.63 | - | 4.51 | [42] |
| Leanhe River, China | Dry season | 7.32 | 259.93 | 5.11 | 25.21 | 1.71 | 0.53 | 1.37 | [43] |
| | Rainy season | 7.19 | 191.07 | 2.19 | 6.99 | 0.35 | 0.07 | 0.99 | [43] |
| Yellow River, China | April | 8.64 | 153 | 5.07 | 6.63 | 0.25 | 0.03 | 5.13 | [40] |
| Yangtze River, China | April | | | 2.86 | 5.40 | 4.69 | 0.96 | - | [12] |
| World average | | | | 1.48 | 0.60 | 0.08 | 0.08 | 0.7 | [44] |
| WHO [a] | | 6.5–8.2 | - | 2000 | 3000 | 10 | 3 | 50 | [45] |
| China MOH [b] | | 6–9 | - | 1000 | 1000 | 50 | 5 | 50 | [46] |
| EQS [c] | I | | | 10 | 50 | 10 | 1 | 10 | [47] |
| EQS [c] | II | | | 1000 | 1000 | 10 | 5 | 50 | [47] |
| EQS [c] | III | | | 1000 | 1000 | 50 | 5 | 50 | [47] |
| EQS [c] | IV | | | 1000 | 2000 | 50 | 5 | 50 | [47] |
| EQS [c] | V | | | 1000 | 2000 | 100 | 10 | 100 | [47] |

Note: [a] Guidelines for drinking water quality, fourth edition (WHO, 2017). [b] Standards for drinking water quality in China (China MOH, 2006). [c] Environmental quality standards for surface water GB 3838-2002 [47].

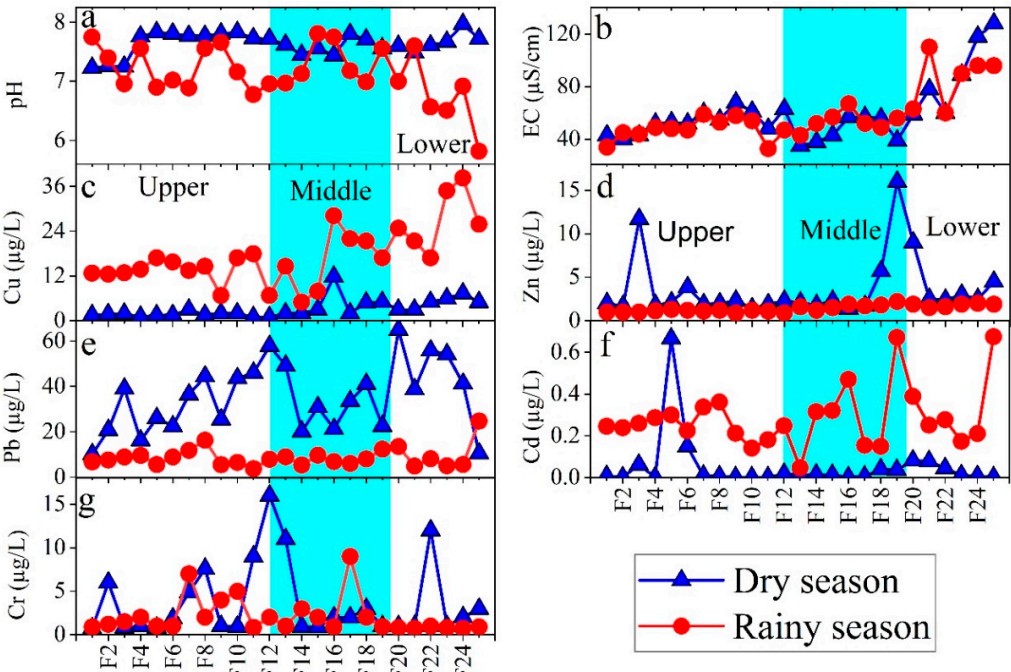

**Figure 3.** Spatial variations of (**a**) pH, (**b**) EC values, and (**c**–**g**) heavy metal concentrations along river flow.

As shown in Figure 2, the concentrations of Zn, Cu, Cd, Pb, and Cr varied greatly. Among all the heavy metals, the concentration of Cd was the lowest (<0.7 μg/L) in the dry and rainy seasons, while the concentration of Pb was the highest in the dry season, varying from 10.11 μg/L to 65 μg/L, and the concentration of Cu was greatest in the rainy season, changing between 5 μg/L and 38 μg/L (Figure 2). The concentrations of heavy metals also showed obvious differences between the two seasons (Figure 2). The concentrations of Cd and Cu in the dry season were lower than those in the rainy season, and they were mainly affected by human activities, such as municipal and industrial wastewater and agricultural production in the rainy season [41,42]. Cr, Pb, and Zn concentrations in the dry season

were greater than those in the rainy season (Figure 2), which may be related to the dilution effect of rainfall in the rainy season [6,43].

Except for Cr, the spatial dispersions and variabilities of other heavy metals in river water were large (Figure 3). The concentration of Cu in river water showed an increasing trend along river flow in both the dry and the rainy seasons, especially in the middle and lower reaches of the river in the rainy season (Figure 3c). The concentration of Zn also tended to increase, with a small amplitude in the rainy and dry seasons (Figure 3d). It was noted that there were two Zn anomalies in the dry season, and anomalies of Cr, Cd, and Pb also occurred along the river flow, which were inseparable from human influence (Figure 3e–g). The Pb concentration increased and oscillated along the river flow in both rainy and dry seasons, showing a larger oscillation, especially in the dry season (Figure 3e). The variation of Cd concentration along the river flow was small in the dry season, except for high abnormal values in the upper reach, and the oscillation amplitude was large in the rainy season, showing an increasing trend along river flow as a whole (Figure 3f). The Cr concentration of river water obviously oscillated in the middle and upper reaches of the river in the rainy season, but it was relatively stable in the lower reach. Except for abnormal values in the upper, middle, and lower reaches of the river in the dry season, there was, overall, a stable variation in Cr concentration along the river flow (Figure 3g).

Compared with other rivers within the Poyang Lake Basin (Table 2), the Pb concentration in the Fu River was greater in the dry and rainy seasons, while the Zn concentration was lower. The Cu concentration of the Fu River was lower than those of other rivers, except the Ganjiang River in the dry season, while the Cu concentration was much greater than those of other rivers in the rainy season (Table 2). In the dry season, the Cr concentration in the Fuhe River was higher than concentrations in other rivers, except for the Xinjiang River, while in the rainy season, the Cr concentration was lower than those in other rivers, except for the Le'an River (Table 2). In addition, the Cd concentration in the Fuhe River was lower than concentrations in the Le'an River in the dry season, and it was greater than that in the Le'an River during the rainy season (Table 2). It was noted that the Cu concentration of the Fuhe River in the rainy season exceeded the surface water quality standard I value of China, and the Pb concentration of this river in the dry season was significantly greater than the surface water quality standard II value of China and the WHO drinking standard value (Table 2). Therefore, the water quality protection of this river should be strengthened.

### 3.2. Influencing Factors of Heavy Metals in River Water

The results of principal component analysis and correlation analysis were shown in Tables 3 and 4, respectively. In order to verify the applicability of the principal component analysis, Kaiser-Meyer-Olkin (KMO) and Bartlett tests were conducted on the data of heavy metal concentrations. In general, the KMO value is >0.5 when the Bartlett's detection significance is $p < 0.05$, indicating that the principal component analysis was effective [41,42]. The KMO and Bartlett (p) test values of the Fuhe River were 0.51 and 0.032, respectively, in the dry season, and they were 0.60 and 0.00, respectively, in the rainy season, showing that the principal component analysis was effective for our data. The variability information of seven variables, exceeding 85.62% and 75.66% in the dry and rainy seasons, respectively, was reflected by four and three principal components, with characteristic values greater than one, respectively, for the Fuhe River (Table 3), indicating that the principal component factor model can explain the variabilities of most variables [40]. In general, absolute load values > 0.75, 0.75–0.5, and 0.5–0.3 are considered as strong, medium, and weak loads, respectively [44].

**Table 3.** Rotated principal component loadings of heavy metal, pH, and EC values in the Fuhe River water.

| Variables | Dry Season | | | | Rainy Season | | |
|---|---|---|---|---|---|---|---|
| | PC1 | PC2 | PC3 | PC4 | PC1 | PC2 | PC3 |
| pH | *0.50* | 0.31 | *0.58* | −0.34 | −0.46 | −0.25 | *0.62* |
| EC | *0.88* | 0.01 | 0.01 | −0.16 | *0.79* | 0.14 | 0.03 |
| Zn | −0.01 | 0.00 | 0.01 | *0.94* | *0.82* | 0.24 | 0.13 |
| Cu | *0.71* | −0.15 | −0.35 | 0.25 | *0.94* | −0.06 | −0.03 |
| Cd | −0.15 | −0.16 | *0.85* | 0.02 | 0.14 | *0.89* | 0.24 |
| Pb | 0.02 | *0.90* | 0.02 | 0.17 | 0.08 | *0.93* | −0.18 |
| Cr | −0.20 | *0.79* | −0.20 | −0.30 | −0. 29 | −0.17 | −0.71 |
| Eigenvalue | 1.75 | 1.60 | 1.29 | 1.02 | 2.9 | 1.45 | 1.0 |
| % of variance | 24.94 | 22.73 | 18.37 | 14.52 | 40.87 | 20.68 | 14.11 |
| Cumulative % | 24.94 | 47.67 | 66.04 | 80.63 | 40.87 | 61.55 | 75.66 |

Note: The load values > 0.50 or <−0.5 in bold italics were considered significant.

**Table 4.** Pearson correlation matrix of pH, EC values, and heavy metal concentrations in the Fuhe River water.

| Dry Season / Rainy Season | pH | EC | Zn | Cu | Cd | Pb | Cr |
|---|---|---|---|---|---|---|---|
| pH | | *0.45* * | −0.25 | −0.03 | 0.18 | 0.24 | 0.09 |
| EC | −0.26 | | −0.12 | *0.41* * | −0.09 | 0.05 | −0.1 |
| Zn | −0.23 | *0.55* * | | 0.07 | 0.02 | 0.05 | −0.22 |
| Cu | −0.38 | *0.64* ** | *0.74* ** | | −0.16 | −0.02 | −0.16 |
| Cd | −0.06 | 0.28 | 0.39 | 0.09 | | −0.08 | −0.15 |
| Pb | −0.34 | 0.19 | 0.25 | 0.06 | *0.69* ** | | *0.49* * |
| Cr | 0.08 | −0.16 | −0.21 | −0.21 | −0.22 | −0.09 | |

Note: The bold italics indicate significant correlations; ** indicates significant correlations at the 0.01 level (bilateral); * indicates significant correlations at the 0.05 level (bilateral).

In the dry season, PC1 explained 24.94% of variance and showed moderate to strong positive loads related to pH, EC, and Cu (Table 3). EC represented the levels of river ion concentrations, reflecting the combination of rock weathering and human activities in the river watershed. The pH value was positively correlated with EC, and Cu was not correlated with other variables (Table 4), indicating that the change in river pH was mainly controlled by rock weathering and human activities, while Cu concentrations were partly derived from rock weathering and human activities. Having commonality, the weathering of watershed rocks leads to a decrease in pH in river water through the consumption of atmospheric $CO_2$, and pollutant discharge from human activities could also cause a similar variation in river water pH because pollutants contain acidic substances [43,48]. In addition, the weathering of silicate rocks (e.g., granite) could release heavy metal ions, such as $Cu^{2+}$, into the river water of the Fuhe River watershed [49]. Wastewater from human activities, such as chemical industries, printing, dyeing, and electroplating, usually contain heavy metal ions, such as $Cu^{2+}$, and they could also cause increasing concentrations of heavy metals in river water once they are released into river water [48]. Therefore, PC1 represented rock weathering and human activities. PC2 explained 22.73% of variance and showed strong positive loads relating to Pb and Cr, which had a significant correlation (Tables 3 and 4), indicating that they had common sources, including inputs from vehicle emissions, as well as wastewater from the mining, smelting, printing, dyeing, and photographic industries [45,47]. PC3 explained 18.37% of the variance and showed moderate to strong positive loads related to pH and Cd (Table 3), but the correlation between the two variables was not significant (Table 4). Cd came mainly from wastewater and waste gas in mining and smelting processes, as well as pesticides and fertilizers in agricultural activities [41,50,51]. Therefore, PC3 represented human activities associated with Cd. PC4

explained 14.52% of the variance, only showing a strong positive load related to Zn, which had no significant correlation with other variables (Tables 3 and 4). Zn is usually believed to come from urban sewage [52], agricultural combustion, or fungicides [53,54], and it may also come from rock weathering [6,55]. Therefore, PC4 indicated that Zn originated from human activities or rock weathering. The result of hierarchical cluster analysis further showed that the Fuhe River watershed was divided into three pollution areas (C1, C2, and C3) during this season (Figure 4). C1 included the midstream sampling points, such as F11, F12, F13, and F22, with maximum Pb and Cr concentrations (Table 5). Industrial activities caused great pollution to river water in the middle reach, where the paper making, smelting, and chemical industries were relatively developed [51,52]. C2 included upstream sampling points, such as F4, F5, F9, and F10, as well as downstream sampling points, such as F17, F20, F21, F23, F24, and F25 (Figure 4). This area was characteristic by a larger EC value (Table 5), representing the most polluted river reaches, whose impact factors were mainly rock weathering and human activities. Of the most polluted river reaches, the upper reach, where there are some industries, such as mining (but having few residents engaging in this industry), could be mainly polluted by mining wastewater and rock weathering, whereas the lower reach was affected by many influences, such as chemical industries, agricultural fertilizers, and resident living with an increasing population (Table 2). C3 included the most upstream sampling points F1, F2, and F3, as well as the sampling points F14, F15, F16, and F19 in the Yihuang tributary in the middle reach of the Fuhe River. This area was represented by the lowest EC value and the highest Zn value (Table 5), reflecting small impacts of human activities, such as urban sewage and agricultural combustion.

In the rainy season, PC1 explained 40.87% of variance and showed strong positive loads related to EC, Zn, and Cu, as well as a medium load related to Cr (Table 3). At the same time, EC, Zn, and Cu had significant correlations among them, but Cr was not correlated with other variables (Table 4). It was thus concluded that Zn and Cu were mainly affected by rock weathering and human activities during the rainy season. The inverse relationships between the loads of EC, Zn, Cu, and Cr indicated that Cr had different sources [6]. Since the natural conditions of the Fuhe River watershed are constant, the strong correlations between EC, Zn, and Cu implied that these heavy metals were mainly affected by human activities in the rainy season [41]. As a result, PC1 represented the combination of rock weathering and human activities in the rainy season, and Cr was related to human activities. PC2 explained 20.68% of the variance, showing strong positive loads related to Pb and Cd which had a significant correlation (Tables 3 and 4). Therefore, Pb and Cd had common sources, including mining and smelting, industrial wastewater, coal combustion, and automobile exhaust emissions [45,47]. PC3 explained 14.11% of the variance, and it only showed a strong positive load related to pH (Table 3). At the same time, pH was negatively correlated with other variables (Table 4), indicating that pH was affected by rock weathering and the buffering effect of hydrochemical ions from human activities [32]. The result of hierarchical cluster analysis further showed that the Fuhe River watershed was divided into two pollution areas (C1 and C2) in this season (Figure 4). C1 and C2 include the sampling points of the middle-upper and lower reaches, respectively (Figure 4). The EC value of C2 was roughly twice as much as that of C1, and the concentrations of all the heavy metals in C2 were higher (Table 5), indicating that there were significant impacts of human activities in the lower reach in the rainy season.

Taken together, there were obvious differences in influencing factors of pH, EC values, and heavy metal concentrations in the Fuhe River in the dry and rainy seasons, which could be related to differential weathering of rocks and different patterns of human activities between the two seasons. At the same time, spatial classifications of heavy metals in river water differed between dry and rainy seasons (Figure 4). The contribution of human activities to river water was relatively high due to small runoff in the dry season, whereas river runoff diluted the input of human activities in the rainy season. In essence, human activities played a significant role in the heavy metal distributions of river water.

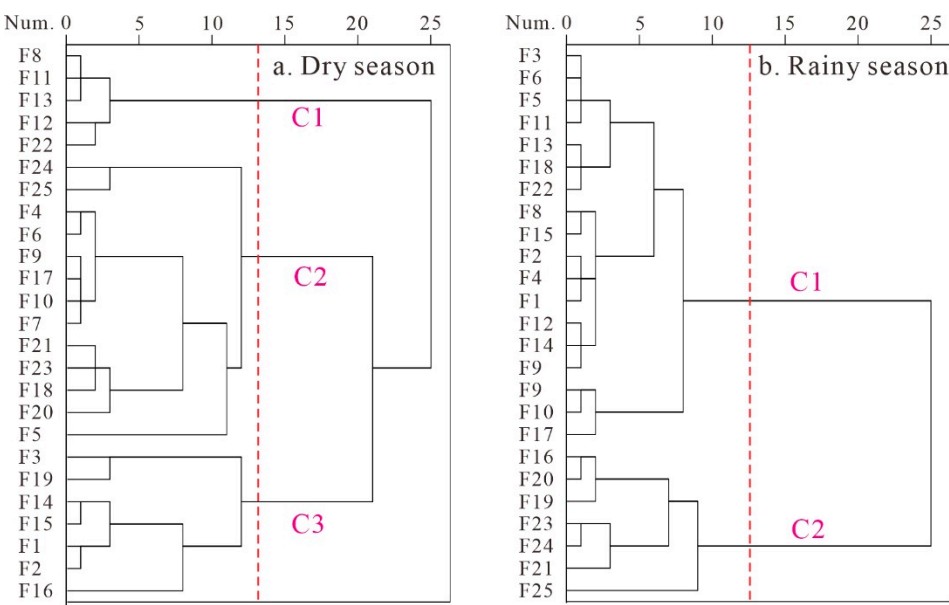

**Figure 4.** Hierarchical cluster tree of sampling sites in the Fuhe River watershed by cluster analysis.

**Table 5.** Mean values with standard deviation (SD) for heavy metals and other parameters in different clusters of the Fuhe River watershed, China.

| Variables | Dry Season | | | | | | Rainy Season | | | |
| | C1 ($n = 5$) | | C2 ($n = 13$) | | C3 ($n = 7$) | | C1 ($n = 18$) | | C2 ($n = 7$) | |
| | Mean | SD | Mean | SD | Mean | SD | Mean | SD | Mean | SD |
|---|---|---|---|---|---|---|---|---|---|---|
| pH | 7.69 [b] | 0.07 | 7.75 [c] | 0.12 | 7.39 [bc] | 0.14 | 7.18 | 0.36 | 7.02 | 0.69 |
| EC | 52.20 | 11.17 | 71.6 [c] | 25.28 | 43.29 [c] | 6.40 | 49.1 | 7.64 | 82.6 | 20.4 |
| Zn | 2.14 | 0.21 | 3.26 | 2.12 | 5.27 | 6.00 | 1.24 | 0.28 | 1.90 | 0.21 |
| Cu | 2.12 | 1.66 | 3.23 | 1.89 | 3.89 | 3.77 | 13.9 | 4.74 | 27.1 | 7.36 |
| Cd | 0.017 | 0.017 | 0.082 | 0.18 | 0.020 | 0.022 | 0.24 | 0.08 | 0.41 | 0.21 |
| Pb | 50.72 [ab] | 5.98 | 34.99 [a] | 15.06 | 23.57 [b] | 9.13 | 8.10 | 2.82 | 10.5 | 7.24 |
| Cr | 11.12 [ab] | 3.22 | 1.81 [a] | 1.20 | 1.78 [b] | 1.91 | 2.58 | 2.29 | 0.84 | 0.08 |

Note: The unit is μS/cm for EC, and it is μg/L for Zn, Cu, Pb, Cd, and Cr. The different letters indicate statistical differences among zones at $p < 0.05$ (a represents C1 and C2, b represents C1 and C3, and c represents C2 and C3).

### 3.3. Human Health Risk Assessment

River water pollution is harmful to human health through direct ingestion and dermal absorption. For example, the impact of Pb may lead to hypertension, lung cancer, gastric cancer, mental and physical retardation, anemia, and spontaneous abortion [56]. High Cu concentration may cause nervous system diseases and liver diseases, and excessive manganese may damage the nervous system and cause IQ deficiency [57]. Long-term exposure to heavy metals may lead to permanent intellectual disability, as well as attention problems [58]. People exposed to heavy metals indeed face cancer and/or non-cancer risks [4,59]. Therefore, the distributions of heavy metals and their risks to human health should be worth high attention.

Five heavy metals (Zn, Cu, Cd, Pb, and Cr) in the Fuhe River water were selected as evaluation factors to calculate the non-carcinogenic health risk index (HQ) caused by direct ingestion and dermal absorption. As shown in Figure 5a and b, the $HQ_{dermal}$ values of Zn, Cu, Cd, Pb and Cr related to adults varied from $5 \times 10^{-9}$ to $8.89 \times 10^{-8}$, from $2.78 \times 10^{-8}$ to $1.06 \times 10^{-6}$, from $6.66 \times 10^{-8}$ to $6.39 \times 10^{-5}$, from $2.97 \times 10^{-6}$ to $5.16 \times 10^{-5}$, and from $1.67 \times 10^{-5}$ to $3.55 \times 10^{-4}$, respectively. The $HQ_{dermal}$ values of Zn, Cu, Cd, Pb, and Cr related to children varied from $7.18 \times 10^{-9}$ to $1.28 \times 10^{-7}$, from $3.99 \times 10^{-8}$ to $1.52 \times 10^{-6}$, from $9.58 \times 10^{-8}$ to $6.46 \times 10^{-5}$, from $4.27 \times 10^{-6}$ to $7.41 \times 10^{-5}$, and from $2.39 \times 10^{-5}$

to $5.11 \times 10^{-4}$, respectively (Figure 5a,b). The $HQ_{ingestion}$ values of Zn, Cu, Cd, Pb and Cr related to adults varied from $2.01 \times 10^{-4}$ to $2.34 \times 10^{-4}$, from $1.1 \times 10^{-3}$ to $4.16 \times 10^{-2}$, from $8.76 \times 10^{-5}$ to $5.91 \times 10^{-2}$, from 0.12 to 2.03, and from 0.01 to 0.23, respectively (Figure 5c,d). The $HQ_{ingestion}$ values of Zn, Cu, Cd, Pb and Cr related to children varied from $1.56 \times 10^{-4}$ to $3.81 \times 10^{-4}$, from $1.3 \times 10^{-3}$ to $4.94 \times 10^{-2}$, from $1.04 \times 10^{-4}$ to $7.02 \times 10^{-2}$, from 0.14 to 2.41, and from 0.01 to 0.28, respectively (Figure 5c,d). The non-carcinogenic risks of Zn, Pb, and Cr related to adults and children by direct ingestion and dermal absorption were smaller in the rainy season than in the dry season (Figure 5), and this was probably related to river runoff dilution in the rainy season. By contrast, the non-carcinogenic risks of Cu and Cd by direct ingestion and dermal absorption were greater in the rainy season than in the dry season (Figure 5), probably caused by the increase in heavy metal-containing sediments and organic substances due to rainfall scouring and the input of industrial wastewater carried by the high-water-level flood in the rainy season. In addition, the average $HQ_{dermal}$ and $HQ_{ingestion}$ values of the five heavy metals to children were all higher than those related to adults in the two seasons (Figure 5), showing that children faced higher non-carcinogenic risks than adults. This was because children's body organs and systems are easily affected by environmental pollution due to their immaturity. The maximum HQ values of the five heavy metals by direct ingestion and dermal absorption were less than 1 in the rainy season, indicating that these heavy metals were not harmful to human health in this period, which was the same case, except for Pb by direct ingestion, in the dry season (Figure 5). The maximum and mean $HQ_{ingestion}$ values of Pb related to adults and children were greater than 1, especially for some sampling points in the middle and lower reaches in the dry season (Figure 5c). This could cause an increase in lead concentration in human blood, further bringing about anemia, cognitive impairment, hearing impairment, vitamin D metabolism disorder, abdominal pain, and other diseases [60,61].

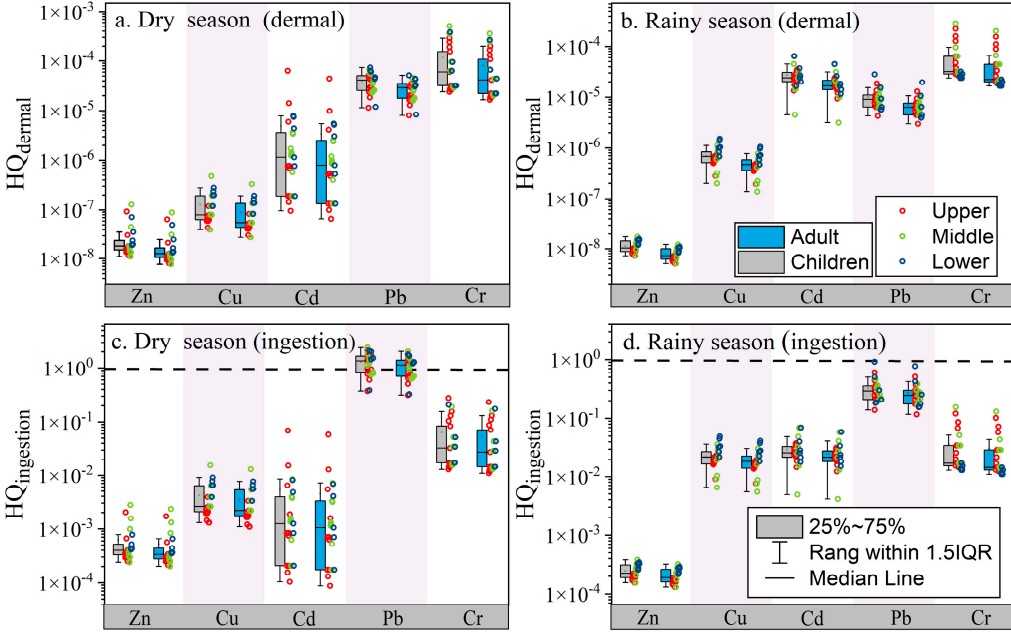

**Figure 5.** Non-carcinogenic risks of Zn, Cu, Cd, Pb, and Cr for children and adults in the Fuhe River watershed, China.

The carcinogenic risk results of Cd and Cr in the Fuhe River water were shown in Figure 6. The $CR_{dermal}$ values of Cd and Cr were, respectively, from $2.03 \times 10^{-12}$ to $1.37 \times 10^{-9}$ and from $1.02 \times 10^{-8}$ to $2.19 \times 10^{-7}$ for adults, and they changed from changed between $2.92 \times 10^{-12}$ and $1.97 \times 10^{-9}$ and from between $1.47 \times 10^{-8}$ and $3.14 \times 10^{-7}$, respectively, for children in the dry and rainy seasons (Figure 6). In the two seasons, the $CR_{ingestion}$ values of Cd and Cr were, respectively, from $2.67 \times 10^{-7}$ to $1.8 \times 10^{-4}$ and

from $1.35 \times 10^{-3}$ to $2.87 \times 10^{-2}$ for adults, and they changed from between $3.17 \times 10^{-7}$ and $2.14 \times 10^{-4}$ and from between $1.6 \times 10^{-3}$ and $3.41 \times 10^{-2}$, respectively, for children (Figure 6). The $CR_{dermal}$ and $CR_{ingestion}$ values of Cd and Cr were, on average, higher for children and for adults. In addition, the $CR_{dermal}$ and $CR_{ingestion}$ values of Cd related to adults and children were lower in the dry season than in the rainy season, but the values of Cr showed inverse seasonality (Figure 6). As shown in Figure 6a, the $CR_{dermal}$ values of Cd and Cr related to adults and children were lower than $10^{-4}$ in the dry and rainy seasons, indicating no harm to human health by dermal absorption. It was noted that the minimum $CR_{ingestion}$ values of Cr related to adults and children by direct ingestion were greater than $10^{-4}$ in the dry and rainy seasons, and the CI values ($CR_{dermal} + CR_{ingestion}$) were also thus greater than $10^{-4}$ (Figure 6b). The results showed that the carcinogenic risks of Cr to adults and children by direct ingestion were significant in the Fuhe River. The higher Cr concentrations in some sampling sites than the Chinese drinking standard value seemed to support the above-mentioned results (Figure 3).

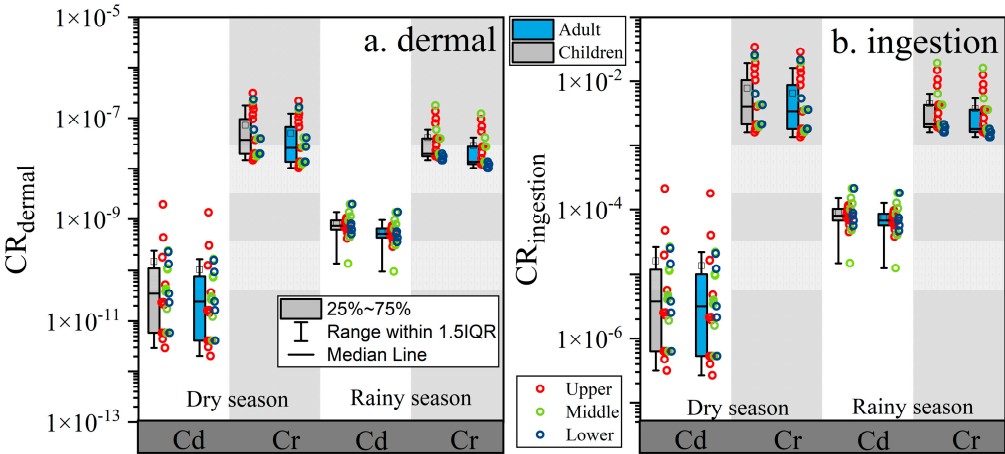

**Figure 6.** Carcinogenic risks of Cd and Cr for children and adults in the Fuhe River watershed, China.

In the rainy season, the $CR_{ingestion}$ value of Cd related to adults was greater than $10^{-4}$ for 16% of the sampling points, and the $CR_{ingestion}$ value of Cd related to children was greater than $10^{-4}$ for 28% of the sampling points, and thus the corresponding CI values of Cd were also greater than $10^{-4}$, and these sampling points were concentrated in the lower reach of the Fuhe River (Figure 6b). For other sampling points, the CI values of Cd related to adults and children were lower than $10^{-4}$ in this period (Figure 6b). These results showed that the carcinogenic risks of Cd related to adults and children were significant in some areas in the rainy season. The $CR_{ingestion}$ values of Cd related to adults and children were greater than $10^{-4}$ only for the sampling point (F5) of the upper reach, while the CI ($CR_{dermal} + CR_{ingestion}$) values of Cd to adults and children were below $10^{-4}$ for other sampling points in the dry season (Figure 6b), showing that potential carcinogenic risks of Cd to adults and children by direct ingestion could only occur somewhere in this period [62].

As a result, in the Fuhe River watershed, it is imperative to strengthen the control and treatment of industrial and mining enterprises' sewage discharge and urban wastewater discharge, especially in the reaches where the concentrations of Pb, Cd, and Cr obviously exceed the Chinese or WHO drinking standards and the human health risk is significant.

## 4. Conclusions

In this study, the spatial variations of pH, EC and Cu, Zn, Pb, Cd, and Cr concentrations in the river water of the Fuhe River watershed in the dry and rainy seasons were investigated in detail, the main influences affecting the concentrations of heavy metals

were explored, and the potential human health risks of heavy metals were evaluated. The main conclusions were obtained as follows.

1. From the upper to lower reaches of the Fuhe River, the pH value of river water changed from between 7.23 and 7.97 in the dry season and from between 5.82 and 7.81 in the rainy season, with a significant decreasing trend along river flow. The EC value of river water increased gradually along river flow in both the dry and rainy seasons. The spatial variations of pH and EC values along river flow were closely related to industrial, agricultural, and urban activities. Especially in the rainy season, the pH decrease and the EC increase in river water in the lower reaches of the Fu River watershed were directly related to frequent human activities.

2. The spatial dispersions and variabilities of heavy metals were large in river water. The average concentrations of heavy metals in the dry and rainy seasons were Pb > Zn > Cr > Cu > Cd in the dry season and Cu > Pb > Cr > Zn > Cd in the rainy season. Except for Pb, the concentration of other heavy metals in the Fuhe River watershed was low, which was superior to the Chinese drinking standard values. The sources of heavy metals seemed to be relatively complex. Overall, Cu and Zn were controlled by natural weathering and human activities, and Pb, Cr, and Cd were mainly from human activities.

3. In the dry and rainy seasons, the HQ values of most heavy metals by direct ingestion and dermal absorption were less than 1. Only in the dry season, especially in the middle and lower reaches of the river, the HQ values of Pb related to adults and children by direct ingestion at most sampling points were greater than 1, indicating significant non-carcinogenic risks for adults and children. In addition, the minimum $CR_{ingestion}$ of Cr in river water was greater than $10^{-4}$ in the dry and rainy seasons. In the rainy season, the $CR_{ingestion}$ value of Cd was $>10^{-4}$ at 16%–28% of the sampling points, and most of them occurred in the lower reach of the river. Therefore, significant cancer risks related to Cr and Cd existed for both adults and children. It is important and imperative to seriously control the inputs of various materials from industries, agricultures, and residential living into river water.

Our finding is helpful for not only understanding the geochemical cycle of heavy metals on a small watershed scale, but also for guiding decision-making on water resource management and water pollution control of the Fuhe River watershed. Nevertheless, there are several aspects of shortcomings for this study. Firstly, two sampling campaigns (January and August) were limited to reflect the whole contamination situation of river water. Secondly, other waterbodies (groundwater, precipitation, and wastewater) were not geochemically considered, but they are hydraulically associated with river water. Finally, the values of toxicity parameters (such as ED, AF, and BW) were mostly based on USEPA recommendations and previous findings [33–39]. However, in fact, these parameters are different from region to region. To completely understand the level, origin, and effect of water pollution for better guidance of surface water quality protection, we suggest that the long-term monitoring should be strengthened for the concentrations of heavy metals in river water in the future studies. Groundwater, precipitation, and wastewater in the watershed should also be investigated in detail. In addition, it is essential to obtain the data of toxicity parameters in the study area.

**Author Contributions:** Conceptualization, X.G. and F.Z.; methodology, Y.X.; software, L.Z.; validation, T.Y. and C.T.; formal analysis, Y.X.; investigation, L.Z.; resources, T.Y. and X.G.; data curation, F.Z.; writing—original draft preparation, X.G.; writing—review and editing, W.L. and C.H.; visualization, C.H.; supervision, F.Z.; project administration, F.Z.; funding acquisition, F.Z. All authors have read and agreed to the published version of the manuscript.

**Funding:** This study was financially supported by the Science and Technique Foundation of the Water Resources Department of Jiangxi Province (Grant Nos. 202123YBKT10) and the Science and Technology Project of Jiangxi Provincial Department of Education (GJJ190971).

**Data Availability Statement:** The authors confirm that the data supporting the findings of this study are available within the article.

**Acknowledgments:** We thank Jingyi Huang for his help in sampling, as well as Wenbo Rao for language editing.

**Conflicts of Interest:** The authors declare no conflict of interest.

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
