# Peer review of "Spatio-Temporal Analysis and Health Risk Assessment of Heavy Metals in Water from the Fuhe River, South China"

_water, doi:10.3390/w15040641_

Round 1
Reviewer 1 Report
The article is aimed at analysing in detail the spatial variations of pH, EC and concentrations of Cu, Zn, Pb, Cd and Cr in the waters of the Fuhe River during the dry and rainy seasons.
Important findings in relation to the chemical characteristics of the river emerge from the discussion, particularly with regard to pH variations during the seasons and the spatial dispersion of heavy metals in the water.
It emerges from the investigation that the lower reaches of the river Fuhe present the highest risks with regard to the possibility of developing cancer in adults and children.
The proposed model also appears interesting with regard to the dynamics of the geochemical cycles of heavy metals in relation to the proposed scale.
The abstract is well articulated, but should be supplemented with information on the results.
Although the proposed subject matter is not highly innovative in terms of the methodologies used, it does stand out positively in relation to the safety of the people living in this context, as well as in relation to the opportunities to develop innovative scientific models concerning the geochemical cycle on a small basin scale.
The entire treatment is coherent and well-developed, which ensures great logicality in the layout of the contribution.
The objectives of the contribution, as well as the methodology used, are well defined and consistent with the subject matter.
The paper is also well-structured, grammatically correct and consistent with the aim of the journal.
The conclusions are well articulated, address the proposed topic in a simple and schematic manner in relation to the analyses carried out.
The subject matter has already been discussed in various works, so there are important and current bibliographical references that guarantee a good contextualisation of the experimentation carried out.
The manuscript is not highly original, but may represent the basis for future experimental and research activities.
It is suggested that Figure 1 be reduced in size and remain within the margins provided in the journal's editorial notes.
Reviewer 2 Report
Article Distributions, Influences and Health Risk Assessment of 2 Heavy Metals in River Water of a Typical Watershed, South 3 China” concerns quality of surface water and its influents on human health”. However, the study are interesting the explanation of research is not sufficient. In general the paper is well written, but some remarks are given below.
2. Study area
Describe more factors affecting surface water quality. The article does not mention the impact of runoff from transportation routes. Both winter road maintenance substances and pollution emitted by traffic can affect the quality of groundwater. What is the distance of sampling points from communication routes? Can road pollution affect river water quality?
There is no information about the classification of surface water quality.
What is the water quality of the catchment against the classification adopted in China?
Does the study conducted in the paper confirm this water quality? It should be written in the section results and discussion.
The section “Samples, analysis and data processing” should be supplemented with more detailed information.
3.1. Sampling and analysis
Please specify the limit of determination, the accuracy of the method and the measurement uncertainty of the elements that have been determined.
3.2. Data processing
3.2.1. Statistical methods
Please write more details about OriginPro2021 and influencing factors of heavy metals in river water with SPSS19.0.
4. Results and discussion
4.2. Influencing factors of heavy metals in river water
Table 3.
Rotated principal component loadings of heavy metals, pH and EC need more explanation.
Please explain how PC1, PC2, PC3, PC4 had been calculated?
Please write more details about how weathering can affect the concentration of heavy metals in waters. Combine this information with the geological structure of the catchment.

Reviewer 3 Report
The authors investigated the presence and health risk assessment of selected heavy metals in a river water in South China. Although the work is important considering the significant health risk of heavy metals and it is also very relevant to the scope of the journal. However, there are some methodological issues that authors need to address or perhaps provide further information before the work can be recommended for publication. The following comments are to be looked into.
Line 54: Replace lags by lag
Line 116: Why was January and August selected? How many times samples were collected in these selected months? For the sample analysis of this nature, replications are required to minimize the errors. Authors should state if the samples were replicated or not.
Line 123: Why are these heavy metal selected?
Line 136-137: Why are some elements selected for non-carcinogenic risk assessment and some for carcinogenic risk assessment? These information should be provide.
There is no any information regarding the principal component analysis and correlation analysis in the methodology. Authors also failed to introduce these analyses and similar work that used these methods. The reasons for selecting them are also not provided.
The samples used to arrive at these conclusions are not sufficient enough.
Reviewer 4 Report
Thank you very much for your text and the appropriate and novel use of the "Factor Analysis" and its ingenious interpretation of each of the components from their loads. Let me just make a minimum remark which I consider can improve the understanding of your text: I believe that you can add the information contained in lines 346 to 354 in the introduction and elaborate a little more on the health effects of these heavy metals in children and adults, which you can return to in paragraph 4.4, but it is necessary to add it from the introduction. E$sto could perhaps be seen as repetitive information, but being so relevant it is necessary to be clear from the beginning
Author Response
Responses to the comments of reviewer 4
Thank you very much for your text and the appropriate and novel use of the "Factor Analysis" and its ingenious interpretation of each of the components from their loads. Let me just make a minimum remark which I consider can improve the understanding of your text: I believe that you can add the information contained in lines 346 to 354 in the introduction and elaborate a little more on the health effects of these heavy metals in children and adults, which you can return to in paragraph 4.4, but it is necessary to add it from the introduction. E$sto could perhaps be seen as repetitive information, but being so relevant it is necessary to be clear from the beginning.
Answer: Thank you very much for your useful comments. According to the comments, the information on health effects of heavy metals was added in the Introduction. Please see lines 49-52 of the revised version.
Reviewer 5 Report
The report by Guo et al. reports on the spatio-temporal variations in the physicochemical quality and health risks associated with ingestion and dermal contact with water from Fuhe River, South China. Justification for the study has been clearly stated, and in my opinion makes the report worth publishing. However, the manuscript needs to be restructured following journal guidelines. I have very little scientific confidence in the health risk assessment results as the reference doses for dermal contact with and direct ingestion of water are not usually the same, unlike assumed in this study. Kindly consider the following suggestions for improvement of the manuscript.
SUGGESTIONS
1. Title
The current title is not very informative. For increased readership, consider revising the title to: Spatio-temporal Analysis and Health Risk Assessment of Heavy Metals in Water from Fuhe River, South China
L14: Delete the second ‘‘Correspondence’’
2. Abstract
L17: quality safety >> security.
-Abbreviations should be written in full at first use.
L18-19: characteristics of pH and EC values, heavy metals concentrations (Cu, Zn, Pb, Cd and Cr) >> physicochemical characteristics and heavy metal (Cu, Zn, Pb, Cd and Cr) content.
Since this is a quantitative study, numerical results (ranges or means with standrad errors) need to be supplied in the abstract.
3. Keywords
Words already in the tile should not be listed as keywords. ‘‘cancer risk’’, could be added as a keyword.
L 114: Indicate what F1-F25 in the map represent.
L91: 2. Study area >> 2. Materials and Methods
Please refer to the journal guidelines on how to format the methodology/different sections of the manuscript.
What quality assurance and quality control measures were taken during the analysis e.g., Reagent and procedural blanks, replication of experiments/analysis?
L129-130: As per your results (L308-332), Hierarchical Cluster Analysis was done. Please indicate.
L158-159: Please recheck these RfD values, and be as specific as possible. The RfD for ingestion are not (always) the same as for dermal contact. See for example https://doi.org/10.1016/j.jenvman.2021.113744 (reference no. [29]) and other reports: https://doi.org/10.1186/s12199-019-0812-x, https://doi.org/10.1186/s13104-020-4939-z . Reference [6] cited did not perform any health risk assessment, and should be omitted.
L245: Having a plot could give a better visualization of these PCA results.
L304: In a word >> Taken together.
L308-344: These should be placed under/merged with Section 4.1 (from L245).
For results on carcinogenic and non-carcinogenic health risks, it is important to at least give the ranges of the values obtained other than just indicating that they did not or did exceed the RfD or the cancer borderlines. I suggest rechecking the RfD values used; it is likely that there are actually no CR from use of water from this river.
Conclusions
Suggestions for future research could be made.
Round 2
Reviewer 2 Report
Review
Many things have been corrected in the revised article, but two still need clarification.
1. Please explain how PC1, PC2, PC3, PC4 had been calculated?
2. Please explain what SPSS means.

Author Response
Comments and Suggestions for Authors
Review
Many things have been corrected in the revised article, but two still need clarification.
Point 1: 1. Please explain how PC1, PC2, PC3, PC4 had been calculated?
Answer: Thank you very much for this comment.
PCA is often used in data reduction to identify common factors (principal components, PCs) that explain most of the variance observed in a large number of manifest variables. The reduced new set of orthogonal (non-corrected) PCs by PCA is arranged in decreasing order of merit. PCA was made with varimax rotation of standardized component loadings for maximizing the variation among the variables under each factor, and those PCs with eigenvalue >1 were kept.
In order to verify the applicability of the principal component analysis, Kaiser-Meyer-Olkin (KMO) and Bartlett tests were conducted on the data of heavy metal concentrations. In general, the KMO value is > 0.5 when the Bartlett’s detection significance is p < 0.05, indicating that the principal component analysis was effective. The KMO and Bartlett (p) test values of the Fuhe River were 0.51 and 0.032, respectively in the dry season, and 0.60 and 0.00, respectively in the rainy season. The variability information of 7 variables exceeding 85.62% and 75.66% in dry and rainy seasons, respectively, was reflected by 4 and 3 principal components with characteristic values greater than 1, respectively for the Fuhe River.
Please see lines 159-166 and 295-308 of the revised version.
Point 2: 2. Please explain what SPSS means.
Answer: Thank you very much for this comment.
SPSS is the abbreviation of Statistical Package for Social Science software, which is a statistical analysis software.

Reviewer 3 Report
The comments have been addressed. I suggest accept.
Author Response
Comments and Suggestions for Authors
The comments have been addressed. I suggest accept.
Answer: Thank you very much!
Reviewer 5 Report
The authors have addressed all my concerns on the previous draft. Thank you
Author Response
Comments and Suggestions for Authors
The authors have addressed all my concerns on the previous draft. Thank you
Answer: Thank you very much!
Reviewer 6 Report
The authors have improved the paper substantially and I think its ready for publication.
Author Response
Comments and Suggestions for Authors
The authors have improved the paper substantially and I think its ready for publication.
Answer: Thank you very much!